# Network-based study of Lagrangian transport and mixing

Kathrin Padberg-Gehle[1] and Christiane Schneide[1]

[1]Leuphana Universität Lüneburg, Institute of Mathematics and its Didactics, Applied Mathematics Group, Universitätsallee 1, D-21335 Lüneburg, Germany

*Correspondence to:* padberg@leuphana.de

**Abstract.** Transport and mixing processes in fluid flows are crucially influenced by coherent structures and the characterization of these Lagrangian objects is a topic of intense current research. While established mathematical approaches such as variational or transfer operator based schemes require full knowledge of the flow field or at least high resolution trajectory data, this information may not be available in applications. Recently, different computational methods have been proposed to identify coherent behavior in flows directly from Lagrangian trajectory data, that is, numerical or measured times series of particle positions in a fluid flow. In this context, spatio-temporal clustering algorithms have been proven to be very effective for the extraction of coherent sets from sparse and possibly incomplete trajectory data. Inspired by these recent approaches, we consider an unweighted, undirected network, where Lagrangian particle trajectories serve as network nodes. A link is established between two nodes if the respective trajectories come close to each other at least once in the course of time. Classical graph concepts are then employed to analyze the resulting network. In particular, local network measures such as the node degree, the average degree of neighboring nodes, and the clustering coefficient serve as indicators of highly mixing regions, whereas spectral graph partitioning schemes allow us to extract coherent sets. The proposed methodology is very fast to run and we demonstrate its applicability in two geophysical flows - the Bickley jet as well as the antarctic stratospheric polar vortex.

## 1 Introduction

The notion of coherence in time-dependent dynamical systems is used to describe mobile sets that do not freely mix with the surrounding regions in phase space. In particular, coherent behavior has a crucial impact on transport and mixing processes in fluid flows. The mathematical definition and numerical study of coherent flow structures has received considerable scientific interest for the last two decades. The proposed methods roughly fall into two different classes, geometric and probabilistic approaches, see Allshouse and Peacock (2015) for a discussion and comparison of different methods. Geometric concepts aim at defining the boundaries between coherent sets, i.e. codimension-1 material surfaces in the flow that can be characterized by variational criteria (see Haller (2015) for a recent review). Central to these constructions is the Cauchy-Green strain tensor, which is derived from the derivative of the flow map. Thus, full knowledge of the flow field or at least high resolution trajectory data is required for these methods to work successfully. This applies also to other geometric concepts such as shape coherence (Ma and Bollt (2014)). Probabilistic methods aim at defining sets that are minimally dispersive while moving with the flow. The main theoretical tools are transfer operators, i.e. linear Markov operators that describe the motion of probability densities under the action of the nonlinear, time-dependent flow. The different constructions in this family of approaches are reviewed

in Froyland and Padberg-Gehle (2014), also highlighting the crucial role of diffusion in this setting. Recently, a dynamic Laplacian framework has been introduced by Froyland (2015), where explicit diffusion is no longer required in the analytical and computational framework. While for this approach fast and accurate algorithms have been developed in Froyland and Junge (2015), the classical transfer operator setting requires the integration of many particle trajectories for the numerical approximation of the infinite dimensional operator. Here again, full knowledge of the underlying dynamical system is needed, which may not be available in applications. Moreover, all discussed approaches assume that the nonautonomous dynamics is represented by a flow map, which, by construction, only considers the starting and end points of each particle trajectory, but neglects the dynamics between the initial and final points in time.

To overcome these problems, different computational methods have been proposed to identify coherent behavior in flows directly from Lagrangian trajectory data, such as obtained from particle tracking algorithms. One of the earliest attempts is the braiding approach proposed by Allshouse and Thiffeault (2012), where trajectories are classified according to their intertwining pattern in space-time. This method is mathematically sound, but computationally demanding and currently restricted to two-dimensional flows. Trajectory-based approaches have also been introduced by Mancho et al. (2013) and Budišić and Mezić (2012). They use time-integrated quantities along trajectories, which again requires knowledge of the underlying dynamical system. Finally, Williams et al. (2015) attempt to reconstruct the transfer operator from limited amount of trajectory data.

Very recently, spatio-temporal clustering algorithms have been proven to be very effective for the extraction of coherent sets from sparse and possibly incomplete trajectory data (see e.g. Froyland and Padberg-Gehle (2015); Hadjighasem et al. (2016); Banisch and Koltai (2017); Schlueter-Kuck and Dabiri (2017)). Here, distance measures between trajectories are used to define groups of trajectories that remain close and/or behave similarly in the time span under investigation. All these methods can deal with sparse and incomplete trajectory data and do respect the dynamics of the entire trajectories, not just the end points. While $c$-means clustering as used by Froyland and Padberg-Gehle (2015) is computationally inexpensive and works well in example systems (see also Allshouse and Peacock (2015)), spectral clustering approaches as in Hadjighasem et al. (2016); Banisch and Koltai (2017); Schlueter-Kuck and Dabiri (2017), appear to be more robust, but require considerable computational effort.

Inspired by these recent approaches, our aim is to design a reliable but computationally inexpensive method for studying coherent behavior as well as mixing processes directly from Lagrangian trajectory data. For this, we consider an unweighted, undirected network, where Lagrangian particle trajectories serve as network nodes. A link is established between two nodes if the respective trajectories come close to each other at least once in the course of time. This construction is similar in spirit to the concept of recurrence networks (see e.g. Donner et al. (2010a)), but here in a spatio-temporal setting. Whereas in recurrence networks, two points on a trajectory or more generally of a time series are linked when they are close, in the present work we consider a whole trajectory as a single entity. We note that also the discretized transfer operator has been viewed and treated as a network, see e.g. Dellnitz and Preis (2003); Dellnitz et al. (2005); Padberg et al. (2009); Lindner and Donner (2017); Ser-Giacomi et al. (2015). The latter used the directed, weighted network to analyse model data of the Mediterranean Sea with the main focus on in- and out-degrees. A different approach is taken in Donges et al. (2009). The authors compute the mutual information matrix $M$ of a climate data set as adjacency matrix $A$ of an undirected and unweighted network. This way they use the betweenness centrality to identify regions of major importance for energy transport.

We use classical graph concepts and algorithms to analyze our trajectory-based undirected and unweighted flow network. Local network measures such as node degrees or clustering coefficients highlight regions of strong or weak mixing. These and other quantities have been considered in previous work on recurrence networks by Donner et al. (2010a, b), where the authors could link network measures to properties of the underlying dynamical system. In a similar fashion, Lindner and Donner (2017) as well as Ser-Giacomi et al. (2015) considered the in- and out-degrees of a weighted, directed network obtained from a discretized transfer operator and found these to highlight hyperbolic regions in the flow. We note that the node degree in our construction exactly corresponds to the trajectory encounter number very recently introduced by Rypina and Pratt (2017), a quantity that measures fluid exchange and thus mixing. Local clustering coefficients can be related to regular behavior, as has also been observed by Rodríguez-Méndez et al. (2017) in the context of transfer-operator based networks.

In addition to considering local network measures, we will apply spectral graph partitioning schemes for the solution of a balanced cut problem (Shi and Malik (2000)). This allows us to efficiently extract coherent sets of the underlying flow, similar in spirit to the approaches proposed in Hadjighasem et al. (2016); Banisch and Koltai (2017), who considered weighted networks, which are constructed based on using different metrics for determining the distance between two trajectories.

The paper is organized as follows. In section 2 we describe our network construction. This is followed by a discussion of network analysis tools in section 3, where we review several, simple local network measures as well as the spectral graph partitioning approach by Shi and Malik (2000). In section 4 we apply the methodology to two different example systems, a Bickley jet as well as the stratospheric polar vortex. We close the paper with a discussion and an outlook on future work.

## 2 Networks of Lagrangian flow trajectories

In the following, we assume that we have $n \in \mathbb{N}$ Lagrangian particle trajectories from a flow simulation or from a particle tracking experiment in physical space $\mathbb{R}^d$, $d = 2$ or 3. In practice, the particle positions may be given at discrete times $\{0, 1, \ldots, T\}$. We denote the trajectories by $x_i$, $i = 1, \ldots, n$, and the particles positions at a certain time instance $t = 0, \ldots, T$ by $x_{i,t} \in \mathbb{R}^d$. We now set up a network in which the trajectories $x_1, \ldots, x_n$ serve as nodes. We link two trajectories if they come $\epsilon$-close to each other at least once in the course of time. Such an undirected, unweighted network is uniquely described by a symmetric adjacency matrix $A \in \{0, 1\}^{n,n}$. In practice, we construct this from the given trajectories by setting

$$A_{ij} = \begin{cases} \max_{0 \leq t \leq T} \chi_{B_\epsilon(x_{i,t})}(x_{j,t}), & i \neq j \\ 0, & i = j \end{cases}, \tag{1}$$

where $\chi_B$ denotes the indicator function of a set $B \subset \mathbb{R}^d$. So $A_{ij} = 1$, that is, a link is established between trajectories $x_i$ and $x_j$, if and only if at one or more time instances $t$, $x_{j,t}$ can be found in an $\epsilon$-ball $B_\epsilon(x_{i,t})$ centered at $x_{i,t}$ and thus the trajectories $x_i$ and $x_j$ have come $\epsilon$-close. In this way, the network encodes in a compact manner how material is transported in the flow – in space and time.

By an appropriate choice of $\epsilon$ one ensures that the network defined by (1) is connected and in this paper we will only consider connected networks. For instance, if the trajectories are initialized on a regular grid, then a natural lower bound to $\epsilon$ would be

the mesh size. In the case that particles are randomly distributed, $\epsilon$ has to be chosen accordingly. We will study different choices of $\epsilon$ in section 4.

Alternatively, the network might be set up by linking the $k$-nearest neighboring trajectories at each time instance for some $k \in \mathbb{N}$. While this allows us to get rid of the problem of a suitable choice of $\epsilon$ it means that we have to choose a reasonable $k$. In two-dimensional systems a natural choice would be $k = 4$ mimicking the five point stencil, similarly $k = 6$ in three-dimensional systems. If trajectories are initialized on a regular grid this choice again ensures that the resulting network is connected. Our own preliminary studies have indicated that this procedure gives very similar results to the $\epsilon$-based definition in (1) but requires slightly longer computational run times. However, as the construction is not symmetric in general, we will not pursue this in the present work.

We note that the network depends on the time interval under consideration. While the study of different time intervals may reveal relevant information about the time scales and other inherent properties of the dynamics, this will not be the focus of our work here.

## 3 Network analysis

Here, we briefly discuss standard analysis concepts for networks (see e.g. Newman (2003)) and relate them to features of the underlying flow. In particular, we will describe how to extract coherent structures by solving a graph partitioning problem, the balanced minimum cut problem as proposed by Shi and Malik (2000) (see also Hadjighasem et al. (2016)).

### 3.1 Degree matrix and graph Laplacian

From the adjacency matrix $A$ one can derive two other important matrices to describe the network. The degree matrix $D$ is a diagonal matrix with $D_{ii} = d_i$ where $d_i$ is the degree of node $x_i$, i.e. $D_{ii} = \sum_{j=1}^{n} A_{ij}$, that is the number of links attached to node $i$. In our setting, $d_i \in \mathbb{N}$, $i = 1, \ldots, n$. By construction, for our network the degree of a node is non-zero, so there are no isolated nodes.

The non-normalized Laplacian is formed by $L = D - A$, where $D$ is the degree matrix and $A$ the adjacency matrix. By the construction of $A$ and $D$, $L$ is symmetric and the entries of $L$ are

$$L_{ij} = \begin{cases} -A_{ij} & i \neq j \\ D_{ii}, & i = j \end{cases}. \tag{2}$$

and thus $L \in \mathbb{Z}^{n,n}$.

The normalized symmetric graph Laplacian $\mathcal{L} \in \mathbb{R}^{n,n}$ is defined as

$$\mathcal{L} = I_n - D^{-\frac{1}{2}} A D^{-\frac{1}{2}}. \tag{3}$$

$\mathcal{L}$ has non-negative real eigenvalues $0 = \lambda_1 \leq \lambda_2 \leq \ldots \leq \lambda_n$. $w_1 = D^{\frac{1}{2}} \mathbf{1}$ is eigenvector to eigenvalue $\lambda_1 = 0$. The other eigenvalues and corresponding eigenvectors can be characterized variationally in terms of the Rayleigh quotient of $\mathcal{L}$. We come back to this in section 3.3.

### 3.2 Local network measures

**Node degree**

The degree of a node encodes how many other nodes are connected to it. In our setting, it measures how many different trajectories come close to the trajectory represented by the respective node and thus it carries information about fluid exchange. The node degree $d$ is encoded in the diagonal elements $d_i = D_{ii}$ of the diagonal degree matrix $D$, with

$$d_i = \sum_j A_{ij}, \quad i = 1, \ldots, n. \tag{4}$$

The node degree $d$ corresponds to the trajectory encounter number as recently introduced by Rypina and Pratt (2017), who also compared this quantity to finite-time Lyapunov exponents and found good agreement in example systems.

**Average degree of neighboring nodes**

Here one considers the average node degree of the neighbors of a node $x_i$, defined as

$$\langle d \rangle_{nn,i} = \frac{\sum_j A_{ij} d_j}{d_i}, \quad i = 1, \ldots, n. \tag{5}$$

Due to the averaging over all neighboring degrees, $\langle d \rangle_{nn}$ tends to be smoother compared to the simple node degree $d$.

Both $d$ and the average degree of neighboring nodes $\langle d \rangle_{nn}$ will be large, when the corresponding trajectory comes $\epsilon$-close to many different other trajectories. In particular in the context of volume-preserving flows, this is only possible when fluid parcels get stretched and folded. Thus, both $d$ and $\langle d \rangle_{nn}$ are expected to be large in mixing regions and can be at least qualitatively related to finite-time Lyapunov exponents, see Donner et al. (2010a); Padberg et al. (2009); Froyland and Padberg-Gehle (2012); Lindner and Donner (2017); Ser-Giacomi et al. (2015) for related studies. However, whereas finite-time Lyapunov exponents measure the overall stretching at the final time, in our construction also all intermediate times are considered. Establishing a formal mathematical link to finite-time Lyapunov exponents is therefore subject to future research.

**Local clustering coefficient**

Here one considers the induced subgraph formed by the vertex $x_i$ under consideration and the vertices incident to it. The local clustering coefficient $C$ indicates how strongly connected this subgraph is by measuring what proportion of the neighbors of $x_i$ are neighbors themselves:

$$C_i i = \frac{\text{\# triangles connected to } x_i}{\text{\# triples centered around } x_i} = \frac{(A^3)_{ii}}{d_i(d_i - 1)}, \quad i = 1, \ldots, n. \tag{6}$$

In the context of recurrence networks, large clustering coefficients have been found to indicate invariant sets of the underlying dynamics (Donner et al. (2010a).) In flow networks obtained from a discretization of the transfer operator large clustering coefficients have been related to periodic behavior (Rodríguez-Méndez et al. (2017)). In the aperiodic time-dependent setting, invariant sets no longer exist, but instead mobile sets, such as vortices, in which the dynamics is regular. In these regions the

dynamics is mainly characterized by rotation and translation. Therefore, in the course of time, trajectories tend to continue interacting with their initial neighbors and encounter only relatively few different trajectories. So the triples and triangles in the network that are due to initial neighborhood (for sufficiently large $\epsilon$), continue to positively influence the value of the clustering coefficient in regular dynamics. A trajectory in a mixing region will be linked to many other trajectories, and due to the underlying stretching and folding, the proportion of triangles is small. Therefore, the local clustering coefficient $C$ is expected to be large for trajectories in regular regions (i.e. for which $d$ or $\langle d \rangle_{nn}$ is small).

The simple local network measures reviewed here depend on the local properties of the network and therefore, of course, on the choice of $\epsilon$. We will study the $\epsilon$-dependence in our numerical studies in section 4. In the context of recurrence networks, the problem of an appropriate choice of $\epsilon$ has been discussed in Donner et al. (2010b). They considered the edge density $\rho(\epsilon) = \frac{2|E(\epsilon)|}{|V|(|V|-1)}$, where $|V|$ denotes the fixed number of vertices and $|E(\epsilon)|$ the $\epsilon$-dependent number of edges of the network. In the literature, values of $\epsilon$ that maximize $\frac{d\rho}{d\epsilon}$ are proposed as optimal choices of $\epsilon$. In the study of Donner et al. (2010b) (see also our own numerical investigations in section 4) however, it has been shown, that such a choice typically results in very dense networks, which no longer encode the local properties of the underlying dynamics. Instead, a limit of $\rho(\epsilon) \leq 0.05$ has been proposed to give reasonable results.

## 3.3 Spectral graph partitioning

Spectral graph partitioning aims at decomposing a network into components with specific properties. In our setting, the network encodes how material is transported by the flow, both in space and time. We are interested in identifying coherent structures in the flow, which are known to be organizers of fluid transport. From a spatio-temporal point of view, coherent sets are formed by trajectories that stay close to each other (Froyland and Padberg-Gehle (2015)) and thus are more tightly connected than others. Such information can be obtained by solving a balanced cut problem of the network (Hadjighasem et al. (2016)).

As outlined above, the normalized symmetric graph Laplacian $\mathcal{L}$ has non-negative real eigenvalues $0 = \lambda_1 \leq \lambda_2 \leq \ldots \leq \lambda_n$. The second smallest eigenvalue $\lambda_2 \geq 0$ is called algebraic connectivity or Fiedler eigenvalue of a graph (Fiedler (1973)). This eigenvalue is non-zero if and only if the network is connected. More generally, the number of connected components of the network appears as the multiplicity of the eigenvalue zero of the Laplacian matrix. If $\lambda_2 > 0$ but very close to zero, then the network is nearly decoupled and the sign structure of the corresponding eigenvector determines two communities in the network (Fiedler cut). If $\lambda_i, . i = 2, \ldots, k$ for some $k < n$ are close to zero and there is a spectral gap between $\lambda_k$ and $\lambda_{k+1}$, then the network is nearly decoupled into $k$ communities. The corresponding eigenvectors $w_2, \ldots, w_k$ carry information about the location of these communities, which can be verified by considering the Rayleigh quotient of the normalized graph Laplacian, as outlined in Shi and Malik (2000). They used this concept to solve a balanced cut problem for defining communities in the network that are characterized by minimum communication between different communities and maximum communication within communities. Such nearly decoupled subgraphs correspond to bundles of trajectories that are internally well connected but only loosely connected to other trajectories. This is indicative of coherent behavior (see also Hadjighasem et al. (2016)). Instead of considering the eigenvalue problem $\mathcal{L}w = \lambda w$, Shi and Malik (2000) propose to solve the equivalent generalized

eigenvalue problem

$$Lv = \lambda Dv. \tag{7}$$

As both $L$ and $D$ are symmetric and have integer entries, eigenvalue problem (7) turns out to be numerically more convenient than the original one. It has the same eigenvalues $0 = \lambda_1 \leq \lambda_2 \leq \ldots \leq \lambda_n$ and the eigenvectors are related by $w_i = D^{\frac{1}{2}} v_i$,

$i = 1, \ldots, n$. In particular, $v_1 = \mathbf{1}$. The number of leading eigenvalues (i.e. eigenvalues close to zero) indicates the number of nearly decoupled subgraphs. An application of a standard $k$-means clustering algorithm (Lloyd (1982)) can then be employed to extract the sets of interest from the corresponding eigenvectors.

## 4 Examples

### 4.1 Bickley jet

As our first example we consider the Bickley jet proposed by Rypina et al. (2007). It is defined by the streamfunction

$$\Psi(x,y,t) = -U_0 L \tanh(y/L) + \sum_{i=1}^{3} A_i U_0 L \operatorname{sech}^2(y/L) \cos(k_i x - \sigma_i t) \tag{8}$$

and it serves as an idealized model of the stratospheric flow. For better comparison, we use same the parameter values as in Hadjighasem et al. (2016), i.e. $U_0 = 5.414$, $A_1 = 0.0075$, $A_2 = 0.15$, $A_3 = 0.3$, $L = 1.770$, $c_1/U_0 = 0.1446$, $c_2/U_0 = 0.205$, $c_3/U_0 = 0.461$, $k_1 = 2/r_e$, $k_2 = 4/r_e$, $k_3 = 6/r_e$ where $r_e = 6.371$ as well as $\sigma_i = c_i k_i$, $i = 1, 2, 3$. Here, we have dropped the

physical units for brevity. The physical assumptions underlying the model equations and the parameters are described in detail in Rypina et al. (2007). For our choice of parameters and when considered on a cylinder, the system exhibits a meandering central jet and three regular vortices on each side of the jet.

Initial conditions are chosen in the domain $M = [0,20[\times[-3,3]$ and are numerically integrated on the time interval $[10,30]$ using a 4th order adaptive Runge-Kutta scheme and periodic boundary conditions in $x$-direction. We output the particle posi-

tions at integer time steps. We also tested finer temporal resolutions and different time intervals, but these did not significantly change our results for this system. We consider two sets of initial conditions, which we will refer to as cases (i) and (ii) in the following:

   (i) 12,200 points from a regular grid on $M$ with grid mesh size 0.1

   (ii) 1,000 random points uniformly distributed on $M$.

For the first high-resolution setting (i) we study different $\epsilon$ from 0.1 to 0.5 (in steps of 0.05), with $\epsilon = 0.1$ corresponding to the distance between neighboring grid points. The different choices of $\epsilon$ result in values for the edge density $\rho(\epsilon)$ between 0.002 and 0.04, which are well within the proposed limit of $\rho(\epsilon) \leq 0.05$ as considered in Donner et al. (2010b). We found no local maximum of $\frac{d\rho}{d\epsilon}$ in this range. For $\epsilon = 0.5$ the resulting network already has about 3 million links, so a possible peak of $\frac{d\rho}{d\epsilon}$ would lie well outside a computationally reasonable range of $\epsilon$.

For the sparse setting (ii), we start with $\epsilon = 0.5$, for which $\rho(\epsilon) = 0.04$. Significantly smaller values of $\epsilon$ did not produce a connected network in this case. A maximum of $\frac{d\rho}{d\epsilon}$ is detected at about $\epsilon = 1.9$, yielding $\rho = 0.45$, which already corresponds to a dense network. So a reasonable range appears to be $\epsilon \in [0.5, 1.9]$.

In Figure 1 the local network measures for case (i) are plotted with respect to the initial conditions. The left column contains the results for $\epsilon = 0.1$, the middle column for $\epsilon = 0.2$ and the right column for $\epsilon = 0.5$. The top row displays the node degree $d$. As expected, $d$ is high in mixing regions, i.e. where trajectories meet many other trajectories and low in the regular regions, i.e. the six vortices and the jet core. Whereas the result for $\epsilon = 0.1$ appears a bit fuzzy, those for $\epsilon = 0.2$ and $\epsilon = 0.5$ are much sharper. The average node degree of neighboring nodes $\langle d \rangle_{nn}$ (middle row) gives a very pronounced indication of regular and mixing flow behavior for small $\epsilon$, but at $\epsilon = 0.5$, the jet core is no longer highlighted by low values of $\langle d \rangle_{nn}$ due to the increased neighborhood over which averages are taken. The bottom row shows the clustering coefficient $C$. For $\epsilon = 0.1$, the vortex cores are characterized by a zero clustering coefficient. This is due to the fact that in this case $\epsilon$ is chosen as the distance between neighboring grid points. However, in this case, two neighbors of a grid point have initially a distance of at least $\epsilon\sqrt{2}$ and therefore in the vortex core region, with its very regular dynamics, the network does not possess any triangles. For all other values of $\epsilon$ studied, the clustering coefficient gives a very clear indication of different dynamical flow regimes, with high values in regular regions as expected.

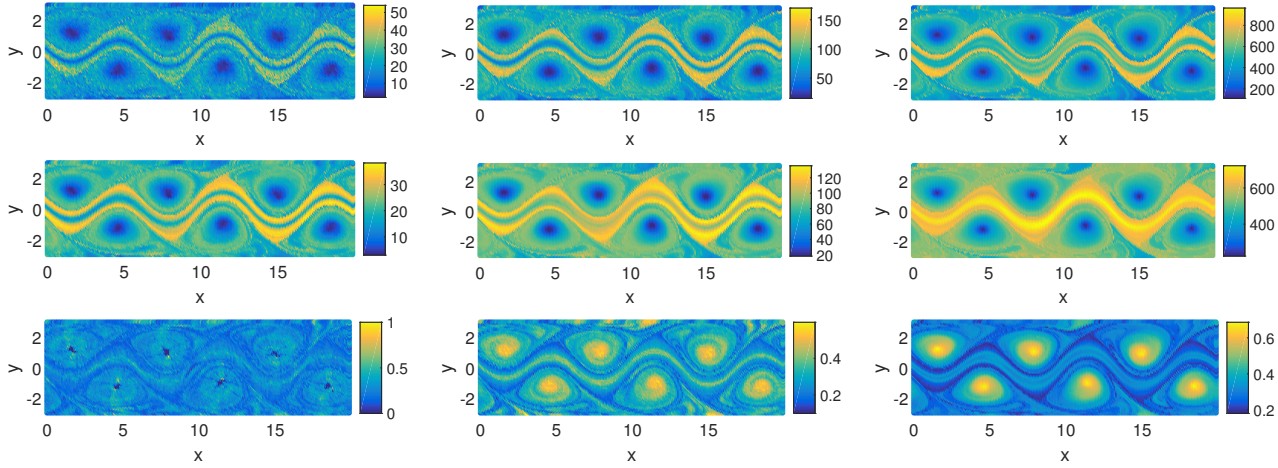

**Figure 1.** Network measures for high resolution initial conditions (case (i)) in the Bickley jet for $\epsilon = 0.1$ (left), $\epsilon = 0.2$ (middle) and $\epsilon = 0.5$. From top to bottom: node degrees $d$ and $\langle d \rangle_{nn}$ and clustering coefficient $C$.

In Figure 2 we repeat the study for the low-resolution case (ii), using $\epsilon = 0.5$ (left column), $\epsilon = 1.5$ (middle) and $\epsilon = 1.9$ (right column). The results are very much comparable to the high-resolution case (i), with the average node degree $\langle d \rangle_{nn}$ (middle row) giving again a good indication of the different flow regimes for small $\epsilon$, where the node degree $d$ only produces spurious results. At $\epsilon = 1.5$, the average node degree $\langle d \rangle_{nn}$ appears to be "switching" and starts to pick up regular regions instead of mixing regions as for smaller $\epsilon$. This is again due to the enlarged neighborhood, where averages are now crucially

influenced by flow regimes outside the local neighborhood of the trajectory under consideration. For instance, for a node with a small node degree, its neighborhood extends far into the mixing regions characterized by large node degrees, resulting in a large average degree for this node (and vice versa for nodes with large node degree). For all choices of $\epsilon$ the local clustering coefficient $C$ picks up the cores of the six vortices, whereas the node degree $d$ is small in these regions and large along the jet, the major transport barrier in this flow. We note that in this sparse setting, the jet core is not resolved by any of the local network measures.

This study supports that the local networks measures are of course $\epsilon$-dependent, but in particular the node degree and the clustering coefficient are robust within a reasonable range of $\epsilon$-values, even for $\epsilon = 1.9$ in the low resolution case. As expected and as found in related work, the clustering coefficient indicates vortices, whereas the node degree highlights major transport barriers. The average node degree $\langle d \rangle_{nn}$ appears to be a good choice for small $\epsilon$, but turns out to be less robust for increasing $\epsilon$, as then larger and larger parts of the network are considered for the averaging and thus the local nature of this network measure decreases.

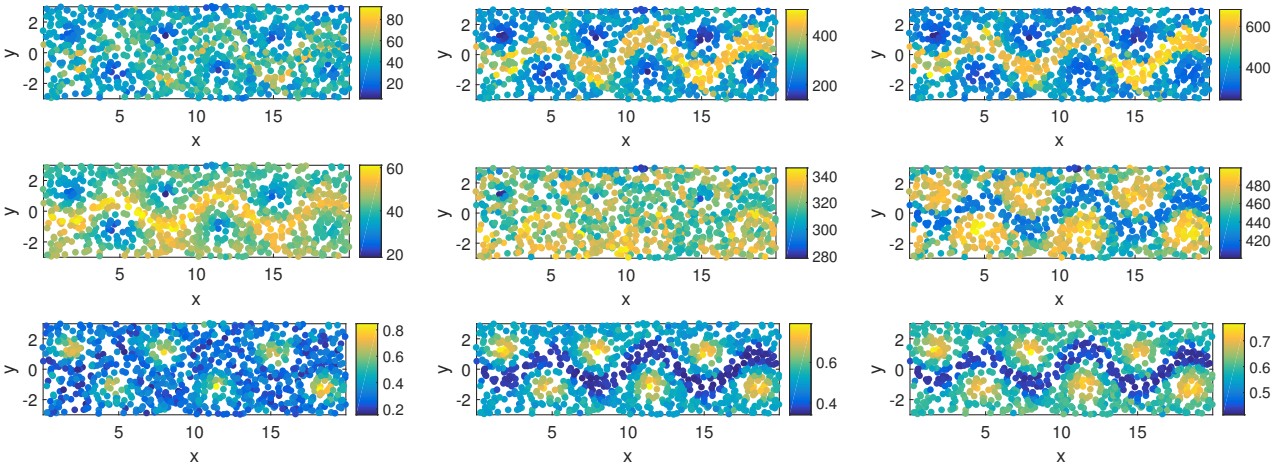

**Figure 2.** Network measures for 1,000 random initial conditions (case (ii)) in the Bickley jet for $\epsilon = 0.5$ (left column), $\epsilon = 1.5$ (middle) and $\epsilon = 1.9$ (right column). From top to bottom: node degrees $d$ and $\langle d \rangle_{nn}$ and clustering coefficient $C$.

In Figure 3 the four (non-trivial) leading eigenvectors $v_2, \ldots, v_5$ of the generalized eigenvalue problem (7) are shown for the high-resolution initial conditions (case (i)) with $\epsilon = 0.2$. The eigenvectors highlight the two regions delineated by the jet as well as the different vortices, comparable to the results in Banisch and Koltai (2017). We note that the corresponding figures for the other choices of $\epsilon$ would look the same. Surprisingly, in the study by Hadjighasem et al. (2016) only the six vortices have been identified but not the different flow regimes delineated by the jet core.

In the low-resolution case (ii), the leading eigenvectors match those of the high-resolution data case, but in a slightly different order (see Figure 4 for the choice $\epsilon = 0.5$). This comes from the fact that the four eigenvalues $\lambda_3, \ldots \lambda_5$ all have approximately the same magnitude and are therefore sensitive to perturbations.

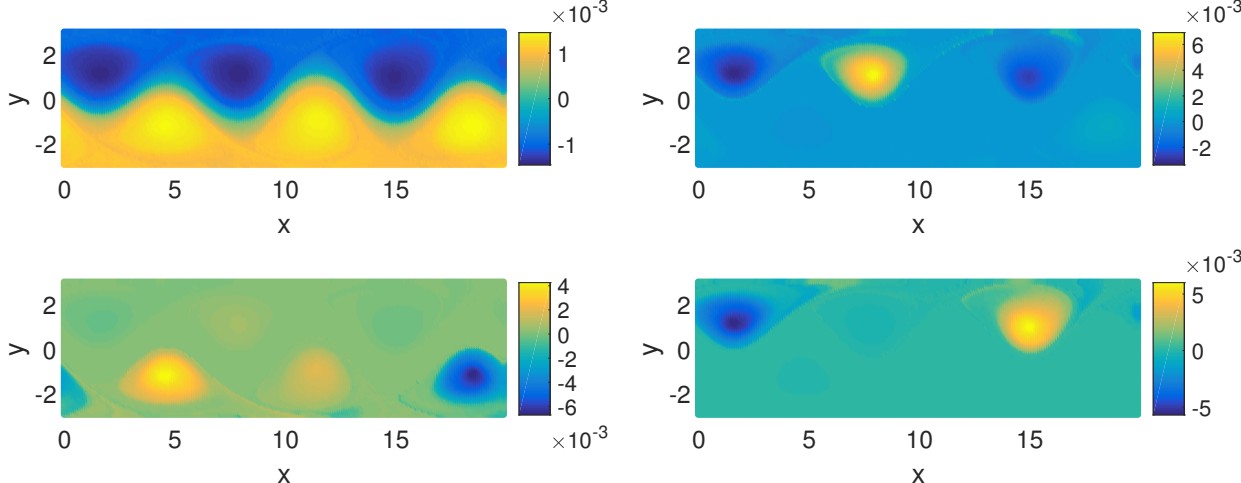

**Figure 3.** Leading eigenvectors $v_2 - v_5$ (from top left to lower right) of the generalized graph Laplacian eigenvalue problem (7) for the network constructed from high resolution initial data in the Bickley jet (case (i)) with $\epsilon = 0.2$.

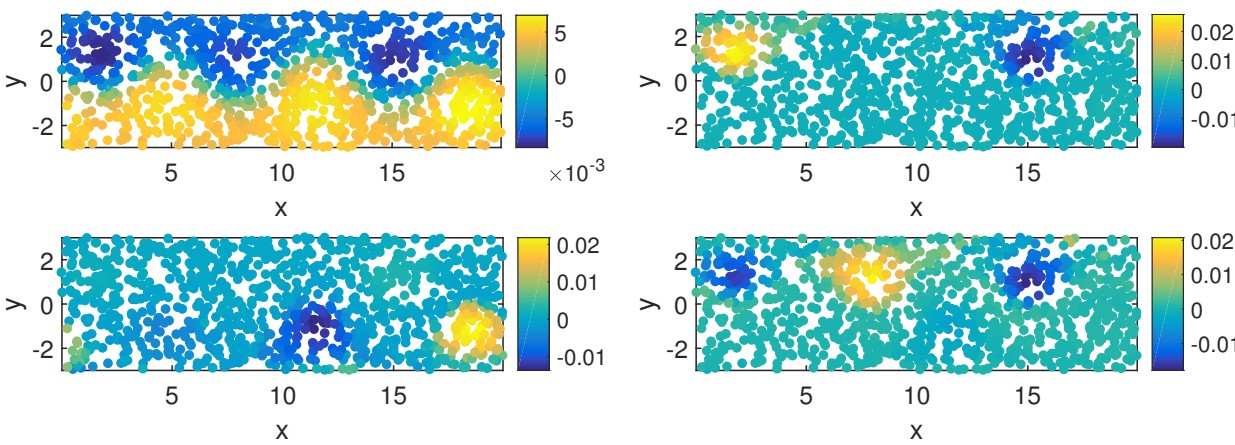

**Figure 4.** Leading eigenvectors $v_2 - v_5$ (from top left to lower right) of the generalized graph Laplacian eigenvalue problem (7) for the network constructed from 1,000 random initial conditions in the Bickley jet (case (ii)) and $\epsilon = 0.5$.

The ten leading eigenvalues for case (i) and $\epsilon = 0.2$ are displayed in Figure 5 (left), the low resolution case (ii) with $\epsilon = 0.5$ in the middle. These spectra exhibit clear spectral gaps between the second and the third and between the eighth and the ninth eigenvalues.

The first spectral gap is related to the coherent behavior of the upper and lower part of the cylinder, delineated by the jet core. The second (and larger) spectral gap indicates the existence of altogether eight coherent sets. These can be extracted via a standard $k$-means clustering (with $k = 8$) of the first eight eigenvectors. The resulting partitions are shown in Figure 6. As

expected, the six vortices and the two distinct stream regions are picked up, both in the high resolution (i) and the sparse data case (ii). However, in the sparse case the clustering finds a few false green and blue points (Figure 6 bottom, left). For the low resolution case (ii) and a choice of $\epsilon = 1$ (or larger) the spectrum is no longer correctly recovered (see Figure 5 (right) for the choice $\epsilon = 1.9$).

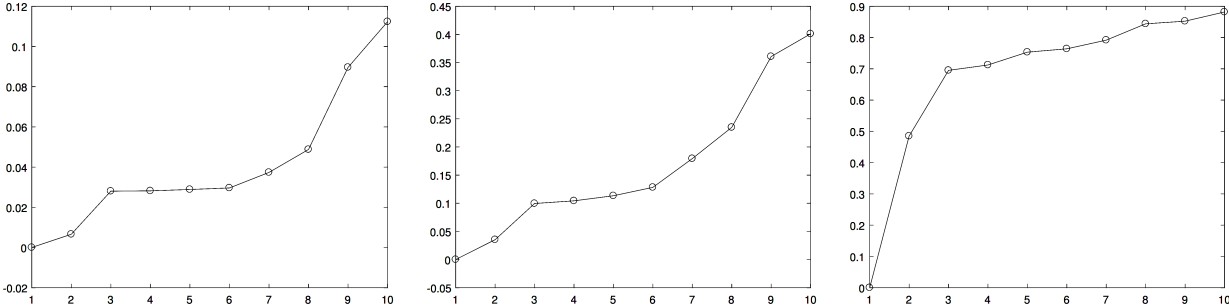

**Figure 5.** Leading eigenvalues of the generalized graph Laplacian eigenvalue problem (7) for the Bickley jet. Left: high-resolution data (case (i), $\epsilon = 0.2$); middle: sparse data case (ii), $\epsilon = 0.5$; right: case (ii), $\epsilon = 1.9$.

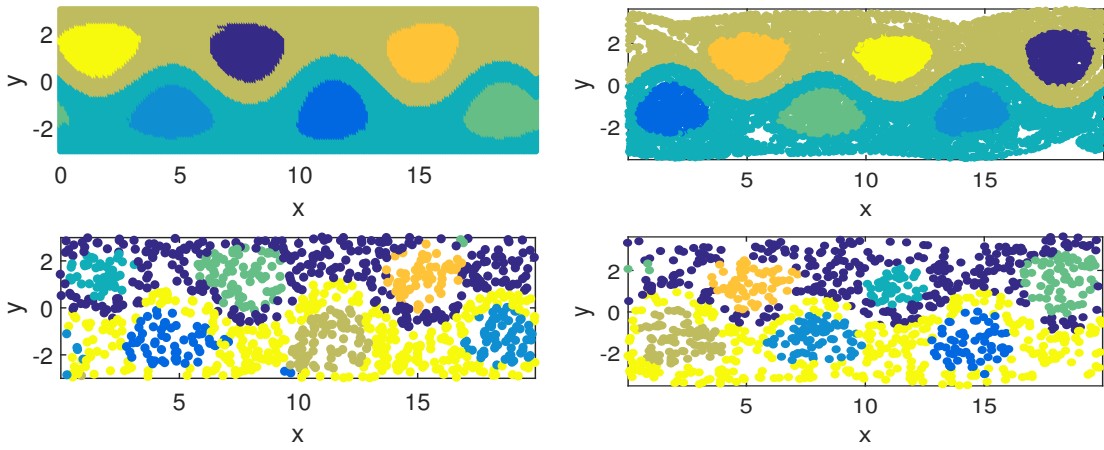

**Figure 6.** Extraction of eight coherent sets based on a $k$-means clustering of the eight leading eigenvectors of the generalized eigenvalue problem for the Bickley jet. Coherent sets at initial time ($t = 10$, left) and at final time ($t = 30$, right). Top: high-resolution case (i), $\epsilon = 0.2$; bottom: 1,000 random initial conditions (case (ii)) with $\epsilon = 0.5$.

5      Finally, we note that the proposed approach is computationally inexpensive, with total run times of $< 2s$ for the sparse data case (ii) and $\approx 40s$ for the high resolution case (i) using MATLAB (R2016a) on a single processor, see Table 1 for details.

**Table 1.** Computation times (in seconds)

|  | trajectory integration | computation of $A$ | eigenvalue problem |
|---|---|---|---|
| (i) 12,200 points ($\epsilon = 0.1$) | 13.4s | 25.9s | 1.8s |
| (i) 1,000 points ($\epsilon = 0.5$) | 1.6s | 0.3s | 0.1s |

## 4.2  Stratospheric polar vortex

As a second example we study the transport and mixing dynamics in the stratospheric polar vortex over Antarctica. The coherent behavior of the polar vortex has already been numerically studied using transfer operator methods (Froyland et al. (2010)). For the computation of particle trajectories we use two-dimensional velocity data from the ECMWF Interim data set[1].

5  The global ECMWF data is given at a temporal resolution of 6 hours and a spatial resolution of a $121 \times 240$ grid in longitude and latitude directions respectively. As in Froyland et al. (2010) we focus on the stratosphere over the southern hemisphere. We consider the flow from September 1, 2002 to October 31, 2002 on a 600 K isentropic surface. For the integration of particle trajectories, we seed initial data on a $64 \times 64$ grid centered at the South Pole (square of side lengths $12,000$km), with a mesh size of $187.5$km. A 4th order Runge Kutta scheme with a constant step size of 45min and linear interpolation in space and

10  time are used and we output the particle positions every six hours. For the construction of the trajectory network we choose $\epsilon = 375$km, i.e. twice the grid spacing. For this choice, we obtain an edge density of $\rho(\epsilon) = 0.03$, which lies well within the reasonable range proposed by Donner et al. (2010b).

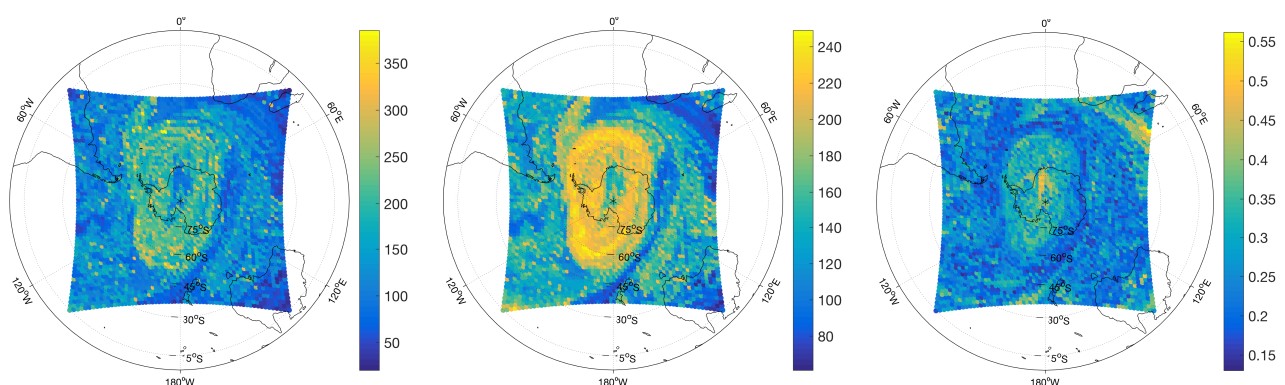

**Figure 7.** Node degree $d$ (left), average node degree of neighboring nodes $\langle d \rangle_{nn}$ (middle) and clustering coefficient $C$ of a network constructed from trajectories for the polar vortex flow between September 1 and October 31, 2002.

In Figure 7 we show the local network measures applied to this network. The node degree $d$ and the average node degree $\langle d \rangle_{nn}$ highlight again the strongly mixing regions that delineate the polar vortex. Similar observations have been made using

---

[1]http://data.ecmwf.int/data/index.html

other stretching measures (see e.g. Joseph and Legras (2002); Froyland and Padberg-Gehle (2012)). The local clustering coefficient is large in particular in the core of the vortex, where the node degree and the average node degree take on small values. As the dynamics is very irregular, the results are less pronounced than in the Bickley jet example, also for larger $\epsilon$ (not shown).

In Figure 8 (top row), the second eigenvector of the generalized graph Laplacian eigenvalue problem (7) is shown. It highlights the polar vortex as a coherent set, confirming the transfer operator based results obtained by Froyland et al. (2010) for a different data set (September 1-14, 2008). However, the result of our computation appears spurious, with small, isolated yellow regions dispersed in the background flow. This is due to a bifurcation in the flow patterns: Towards the end of September 2002, the polar vortex splits into two vortices. One of the two vortices becomes unstable and disperses whereas the other vortex has increased again by the end of the computation (October 31, 2002; see Figure 8 (top, right)). It would be very interesting to identify a precursor of the vortex splitting from the network properties, but this will be subject to future work.

We repeat the study of the spectrum by considering a new network where the trajectories are restricted to the time span before the bifurcation (September 1–26, 2002), see Figure 8 (bottom row). On this interval, the polar vortex can be clearly identified by the second eigenvector of the generalized graph Laplacian eigenvalue problem.

## 5 Discussion and conclusion

We have proposed a very simple and inexpensive approach for analyzing coherent behavior and thus transport and mixing phenomena in flows. It is based on a network in which Lagrangian particle trajectories form the nodes. A link is established between two nodes if the respective trajectories come close to each other at least once in the course of time. The resulting network is unweighted and undirected and can be represented by a binary adjacency matrix. Classical local network measures such as node degree and clustering coefficient highlight regions of strong mixing and regular motion, respectively. While these network measures are $\epsilon$-dependent, they appear to be robust within a reasonable range of $\epsilon$-values. Too large $\epsilon$'s blur the local information of the underlying dynamics and an edge density dependent choice of $\epsilon$ as discussed in the context of recurrence networks (Donner et al. (2010b)) has turned out to be useful in our setting as well. In addition, we have used a generalized graph Laplacian eigenvalue problem to efficiently and robustly extract coherent sets, even for the case of sparse data as illustrated by case (ii) in the Bickley jet investigations.

While in this manuscript we have only demonstrated our approach in examples that are volume-preserving and two-dimensional, the extensions to three-dimensional flows and also to dissipative systems are straight-forward. In addition, although not illustrated here, our method can easily deal with incomplete trajectory data as only one-time encounters of trajectories are required for setting up the network. The approach is not restricted to connected networks, and in particular in the presence of attracting sets in non-volume-preserving systems, these might be worthwhile to consider as well. We have studied unweighted networks throughout the paper. Counting the number of times a trajectory comes close to another is one option for choosing weights. Our own preliminary studies indicate that in this case the node degree and average node degree become less meaningful, as these cannot distinguish any more between repeated encounters (as in regular regions) and many different

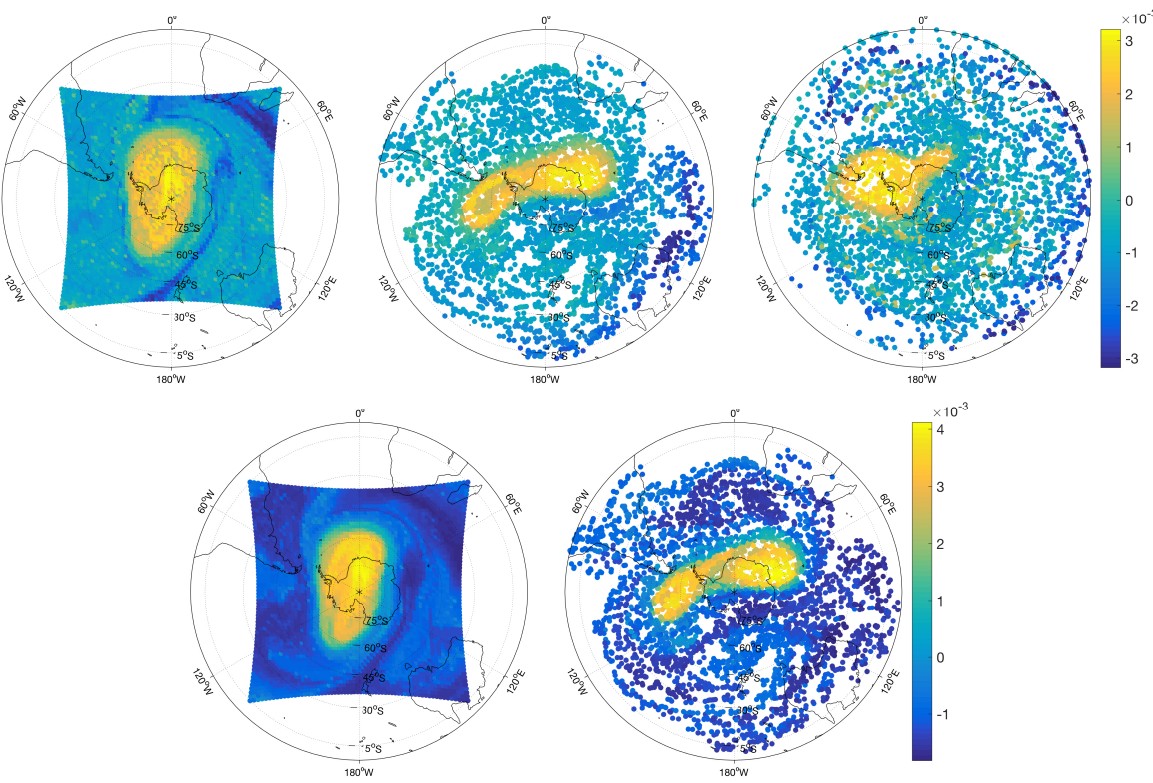

**Figure 8.** Eigenvector $v_2$ of generalized graph Laplacian eigenvalue problem for the polar vortex example from trajectories computed on two different time spans (top: September 1 to October 31, 2002; bottom September 1–26, 2002). Top row: particles at September 1, 2002 (left); September 26, 2002 (middle); October 31, 2002 (right). Bottom row: particles at September 1, 2002 (left); September 26, 2002 (right).

encounters (as in mixing regions). Clustering coefficient and subdominant eigenvectors of the Laplacian appear to continue to highlight coherent regions.

There are some direct relations to other recently proposed methodologies such as the dynamic isoperimetry framework introduced by Froyland (2015), where a dynamic Laplacian and its spectrum play a central role. The graph Laplacian matrix
5 studied in the present paper appears to be a very coarse but inexpensive and robust approximation of this operator and in a similar way it approximates the diffusion maps studied in Banisch and Koltai (2017). In this context, it might be interesting to analyze the networks resulting from the different choices of metrics used in Banisch and Koltai (2017); Schlueter-Kuck and Dabiri (2017); Hadjighasem et al. (2016). A mathematical analysis of the commonalities and differences between these approaches and our novel network approach is subject to future research. Finally, the node degree of our network construction
10 exactly corresponds to the trajectory encounter number very recently introduced by Rypina and Pratt (2017), which has now obtained a wider interpretation in the context of flow networks.

*Acknowledgements.* We thank Gábor Drótos and the second anonymous reviewer for insightful comments and suggestions that helped improve and clarify this manuscript. This work is supported by the Priority Programme SPP 1881 Turbulent Superstructures of the Deutsche Forschungsgemeinschaft (PA 1972/3-1). KPG also acknowledges funding from EU Marie-Skłodowska-Curie ITN Critical Transitions in Complex Systems (H2020-MSCA-2014-ITN 643073 CRITICS). We thank Naratip Santitissadeekorn for sharing code for the Antartic polar vortex computations. Publication is supported by Office of Naval Research under Grant No. N00014-16-1-2492 .

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
