# Peer review of "Network-based study of Lagrangian transport and mixing"

_Nonlinear Processes in Geophysics, 2017_

## Referee Comment (RC1) · Anonymous Referee #1 · 4 Mar 2017

The authors of the above manuscript present a methodology for the characterization of flow systems from a Network Theory perspective. They introduce an unweigthed, undirected network where nodes are represented by Lagrangian particles and links are established when the distance among pairs of particle trajectories results to be smaller of a given threshold $\varepsilon$ during the considered time interval. Then, local measures such as degree, average neighbors degree and clustering coefficient are computed. A spectral partitioning algorithm is also used to find network clusters using the graph Laplacian associated to the adjacency matrix of the network. Such methodology is applied to an ideal flow system, the Bickley jet, and to a realistic one describing the Antarctic stratospheric circulation.

The general topic of the paper is interesting; indeed Network Theory tools are becoming increasingly popular in many research fields, including Geophysical Flows. The

approach here proposed demonstrates its numerical efficiency and broad applicability also to "dirty" datasets and introduce measures that could be useful for further applications to flow systems. However, the method of construction of the network, as it is presented in the paper, does not address the problem of its robustness. This point is extremely relevant because it would affect the reliability of future possible, applications to realistic cases. Moreover, the authors do not discuss similarities and differences of their study with other, relevant, network approaches to dynamical systems that showed up in the last decade. Finally, the authors do not provide solid theoretical foundations for the measures introduced in the paper giving only visual interpretations, mainly steered by the previous knowledge of the systems studied.

Overall, I think this manuscript needs a extended revision to address all the below points before reaching the standard for publication in Nonlinear Processes in Geophysics.

MAJOR POINTS:

– The authors do not provide a proper study of the robustness of their networks to the choice of the crucial parameter $\varepsilon$. They only use 4 different values and plot the results that, in addition, seem to be really sensitive to such parameter. A proper robustness study of $\varepsilon$ is necessary, exploring a larger set of values and understanding the reasons behind the sensitivity of the network to this threshold. About this issue, I warmly suggest the reading of the paper: "Ambiguities in recurrence-based complex network representations of time series" by R. Donner et al. in which the authors deal properly with such thresholding ambiguities.

– The methodology here proposed is really similar to the one presented by A. Hadjighasem et al. in "Spectral-clustering approach to Lagrangian vortex detection" and in some sense is an unweigthed version of their approach. The authors should explain more in detail the differences with the approach of Hadjighasem stating which are the advantages and the disadvantages of their method. An explicit numerical comparison

of the two ways of building the network would significantly improve the quality of the manuscript.

– In general, the authors do not introduce correctly their work in the field of Network Theory application to Geophysical Systems. Climate Networks are not even cited (e.g. the seminal paper by J.F. Donges et al. "The backbone of the climate network" ) and Recurrence Networks are cited superficially without exploring the connections of the method here proposed with the Dynamical Systems Theory and the "recurrence" concept. Moreover, the authors do not confront their methodology with other network approaches previously proposed for the study of Lagrangian transport in flow systems (e.g. E. Ser-Giacomi et al. "Flow networks: A characterization of geophysical fluid transport" and M. Lindner et al. "Spatio-temporal organization of dynamics in a two-dimensional periodically driven vortex flow: a Lagrangian flow network perspective") where the local measure that they used were studied for the first time. In summary, the paper presents not only a lack of citations but also a weak framing of its scientific contents with respect to other previous works. The introduction and the Discussion and conclusions sections require also to be revised in the light of the above.

– The local measures proposed in the paper i.e. degree and clustering coefficient show an interesting detection power for mixing and no-mixing regions. However, this point is not well developed from a theoretical point of view. Which are, for instance, the quantitative relations among the degree centrality in this kind of networks and the classical Lagrangian diagnostics? Why the clustering coefficient seems to detect region of weak mixing? Maybe it could be related to some retention propriety of the flow? What is the meaning of a spectral analysis performed on a adjacency matrix that has been built thresholding Lagrangian trajectories distances at different times? Addressing these questions would certainly increase the impact of this paper, making possible connections with more theoretical approaches to Dynamical Systems.

SPECIFIC COMMENTS:

– Abstract (line 5): not clear, "Lagrangian trajectory data" is a vague definition. Please, specify that the author refer to low-resolution or incomplete data.

– Abstract (line 10): average neighbors degree and clustering coefficient are missing from the list of local network measures, please complete.

– Introduction (line 26): the author should compare here their approach with the work by A. Hadjighasem et al. here.

– Introduction (line 31-32): node degree and clustering coefficient have been already studied in the context of Lagrangian transport (E. Ser-Giacomi et al. "Flow networks: A characterization of geophysical fluid transport" and M. Lindner et al. "Spatio-temporal organization of dynamics in a two-dimensional periodically driven vortex flow: a Lagrangian flow network perspective" and V. Rodriguez-Mendez "Clustering coefficient and periodic orbits in flow networks"). The authors should discuss differences and similarities of their way to compute these metrics comparing with the ones defined previously in the literature.

– Network of Lagrangian flow trajectories (line 9): please reformulate, the expression "network of these Lagrangian trajectories" is confusing.

– Local network measures (line 17): "immediate importance" is not a proper definition for the degree centrality.

– Local network measures (line 26-27): again, the authors should refer to the papers that introduced these kind of network measures previously and not only add self-citations.

– Local clustering coefficient (line 2): what the authors mean for "certain subgraph"?

– Spectral graph partitioning (line 10-13): connected components and communities are not the same thing, please reformulate.

– Bickley jet (line 16-17): as wrote in major points: a study of robustness is needed.

[Figure]

See: "Ambiguities in recurrence-based complex network representations of time series" by R. Donner et al.

– Bickley jet (line 23-24): again, the strong sensitivity of the clustering coefficient to the choice of the threshold demonstrates that a deeper study is needed to address the robustness issue. Note that the clustering pattern is almost inverted just switching from $\varepsilon$=0.1 to $\varepsilon$=0.2.

– Bickley jet (line 29): It is interesting that the clustering seems to be more robust in the low-resolution case, the authors could discuss this point and try to explain the reason.

– Stratospheric polar vortex (line 11): Here the authors use just one value of the threshold $\varepsilon$. It is not sufficient. I think that, if they would perform a systematic robustness analysis for the idealized flow, here it would not needed to repeat it. But, at least, they should show the results for a few values of $\varepsilon$, possibly selected accordingly to the indications from the robustness analysis for the Bickley jet.

– Stratospheric polar vortex (line 12-14): The authors, for consistency, should show the results of the calculation of the clustering coefficient also for this more realistic case.

---

## Referee Comment (RC2) · G. Drotos (Referee) · 24 Mar 2017

The paper entitled "Network-based study of Lagrangian transport and mixing" by Kathrin Padberg-Gehle and Chrisiane Schneide presents a method for identifying mixing regions and coherent sets in fluid flows. It constructs an undirected and unweighted network among trajectories of fluid elements: links represent at least one close encounter between the corresponding trajectories. Classical network measures and graph partitioning schemes are then applied. The method is tested numerically in 2D flows with pronounced vortical features: in a conceptual geophysical model and in atmospheric reanalysis data.

The method introduced in this paper is novel (in particular, it is essentially different from other network approaches that have appeared so far in the literature of fluid dynamics), and it may represent a significant technical advancement in the topic of mixing and coherence in fluid flows. The assumptions underlying the method are plausible, and the results of the numerical tests are convincing about its performance. Its main advantages are its low computational cost and its applicability for sparse or incomplete trajectory data.

The paper is well-written and relatively short, and I find the very concise formulation to be an advantage. There are a few points, however, where giving further details or providing a longer, more general discussion would be useful or even necessary. In certain cases, even further numerical work may be needed.

Based on the above aspects, I definitely recommend to accept the paper, with certain amendments I list below.

Crucial observations:

- An important aspect is the dependence of the constructed network and of the extracted features on the time interval considered. This issue should be discussed. I am thinking of some discussion similar to those in Sections I.b. and V. of Rypina and Pratt (2016). Furthermore, it might also be explained shortly for the example of Bickley jet what would happen if shorter or longer intervals of time were taken. What is more important, however, is a deeper analysis of the case of the stratospheric polar vortex from this point of view. In particular, after one of the two vortices dissolves, the particles formerly composing the dissolving vortex are dispersed within the "background sea", i.e., the larger region around the polar vortex that constitutes a different flow regime. As a result, this "background sea" is "polluted" by light yellow particles in the right plots of Figs. 7 and 8. It should be explained (and maybe numerically confirmed) that this would not happen if the network were constructed from trajectory data only from late October. I would also be happy to see the plots when the network is constructed from trajectory data from 1st to 26th September: maybe the polar vortex is delineated more clearly with this choice, with less green colour seen in the "background" regime. Not

directly related to these considerations, I would be interested as well to know if the dissolution itself of one of the two vortices can explicitly be detected by some network measure (along with some appropriate choice of the time interval)?

- The relation of the network measures applied in this paper to the trajectory encounter number of Rypina and Pratt (2016) is described on lines 30-32 of page 2, 24-26 of page 4, and 20-21 of page 11 as "capturing very similar information, "being very related", or that "there is an obvious link". In my understanding, the trajectory encounter number of Rypina and Pratt (2016) is _exactly_ the same as the node degree d_i introduced in Section 3.2 of the present paper. The difference is that the definition here is expressed in a more formal way, and, more importantly, this quantifier is placed here in a much wider context (that of the trajectory-based flow network). I think that this has to be formulated clearly at each occurrence (enumerated above), and references to other network measures (average degree of neighbouring nodes, local clustering coefficient) have to be completely avoided in this relation.

Further specific points:

Page 4, line 4: "By construction": It would be better to write: "By our choice".

In the same line: "so there are": It should be "since there are" or "i.e., there are".

Page 5, line 13: A period is missing after the parentheses.

Page 5, lines 15-16: A reference should be included for the statement.

Page 5, line 25: Maybe a reference can be included for the k-means as well.

At the beginning of Section 4.1: I understand that the Bickley jet is just an arbitrary example, and the main motivation for choosing it was, I guess, the presence of results in the literature that are comparable to those in the current paper. Still, it would be useful to mention the geophysical relevance of the Bickley jet (especially its relation to the other example: the stratospheric polar vortex), and the main properties of the flow with the current parameter setting (like the presence of a meandering central jet and of

three regular vortices on each side).

Page 6, line 10, and also page 10, line 10: "we output the particle positions ...": Are only these positions considered for the network construction? If so, it should be explicitly stated (and this raises further, maybe crucial questions about the dependence on the choice of the output). If not (as I suppose), where is advantage made of this special filtering of the trajectories in the paper? (For e.g. the interpretation of the figures, informing the reader about this choice of outputting is not needed.)

Page 6, lines 21 and 28: "Spurious" is not an appropriate word here (it approximately means "false"). "Fuzzy" may be a good substitution.

Page 6, line 24: What does "singularity" mean in the present context? Some more explanation should be given.

Page 8, line 5: "four eigenvalues" should be "three eigenvalues".

Figure 6: It might be mentioned very shortly that the clustering finds a few false yellow points for the sparse case.

Page 11, line 14: After the expression "sparse data", it may be emphasized that this is illustrated by case (ii) in the Bickley jet investigation.

Additional general considerations:

- I believe that the applicability of the proposed method is not restricted to volume-preserving flows. For example, this method might be especially useful in open contracting dynamics. It might be beneficial to mention this in the paper, and to indicate that the numerical examples discussed in this paper originate from volume-preserving flows.

- I think that choosing epsilon such that the graph is not connected may also give insightful information in certain cases, especially in non-volume-preserving dynamics, which could be mentioned. Anyway, it should be more clearly stated in line 16 of page

3 that the present paper considers choices of epsilon only that give a connected graph.

- Would the authors consider to be a useful possibility to count the total number of encounters between trajectories occurring in the course of time, and to construct a weighted network based on these numbers? If so, it might be mentioned in Section 5, as a natural extension to the presently proposed method.

Gabor Drotos

---

## Author Comment (AC1) · 13 Jul 2017

**Response to referee reports on "Network-based study of Lagrangian transport and mixing" by Kathrin Padberg-Gehle and Christiane Schneide**

We thank the referees for their careful reading of the manuscript and their very insightful and constructive comments and suggestions and apologize for the delayed response.

We respond to all criticism below and point out the changes in the manuscript we propose to make in order to meet the referees' requests.

**Referee # 1**

1. *The authors of the above manuscript present a methodology for the characterization of flow systems from a Network Theory perspective. They introduce an unweigthed, undirected network where nodes are represented by Lagrangian particles and links are established when the distance among pairs of particle trajectories results to be smaller of a given threshold ? during the considered time interval. Then, local measures such as degree, average neighbors degree and clustering coefficient are computed. A spectral partitioning algorithm is also used to find network clusters using the graph Laplacian associated to the adjacency matrix of the network. Such methodology is applied to an ideal flow system, the Bickley jet, and to a realistic one describing the Antarctic stratospheric circulation. The general topic of the paper is interesting; indeed Network Theory tools are becoming increasingly popular in many research fields, including Geophysical Flows. The approach here proposed demonstrates its numerical efficiency and broad applicability also to "dirty" datasets and introduce measures that could be useful for further applications to flow systems. However, the method of construction of the network, as it is presented in the paper, does not address the problem of its robustness. This point is extremely relevant because it would affect the reliability of future possible, applications to realistic cases. Moreover, the authors do not discuss similarities and differences of their study with other, relevant, network approaches to dynamical systems that showed up in the last decade. Finally, the authors do not provide solid theoretical foundations for the measures introduced in the paper giving only visual interpretations, mainly steered by the previous knowledge of the systems studied. Overall, I think this manuscript needs a extended revision to address all the below points before reaching the standard for publication in Nonlinear Processes in Geophysics.*

   We thank the referee for their careful reading of the manuscript and for their very helpful comments and suggestions. We address these below.

2. *The authors do not provide a proper study of the robustness of their networks to the choice of the crucial parameter $\epsilon$. They only use 4 different values and plot the results that, in addition, seem to be really sensitive to such parameter. A proper robustness study of $\epsilon$ is necessary, exploring a larger set of values and understanding the reasons behind the sensitivity of the network to this threshold. About this issue, I warmly suggest the reading of the paper: "Ambiguities in recurrence-based complex network representations of time series" by R. Donner et al. in which the authors deal properly with such thresholding ambiguities.*

   The referee is right and we have extended the robustness study and dealt with the sensitivity issue in the light of the results in the paper "Ambiguities in recurrence-based complex network

representations of time series" by Donner et al. as suggested above. In particular, we have considered the following:

- We have studied $\epsilon = 0.1$ to $\epsilon = 0.5$ (in steps of 0.05) for the Bickley jet, high resolution case (i) and propose to show results for $\epsilon = 0.1, 0.2$ and 0.5 in the revised paper, see Figure 1 below. For this setting, we have also studied the $\epsilon$-dependence of the edge density $\rho(\epsilon) = \frac{2|E(\epsilon)|}{|V|(|V|-1)}$, where $|V|$ denotes the fixed number of vertices and $|E(\epsilon)|$ the $\epsilon$-dependent number of edges. We found no local maximum of $\frac{d\rho}{d\epsilon}$ in this range. For $\epsilon = 0.5$ the resulting network already has about 3 million links, so a possible peak of $\frac{d\rho}{d\epsilon}$ would lie well outside a reasonable range of $\epsilon$. In Donner et al. $\rho \leq 0.05$ is proposed as an appropriate choice for analyzing local flow structures, the chosen range of $\epsilon = 0.1$ to $\epsilon = 0.5$ gives rise to values of $\rho = 0.002$ to 0.04, satisfying this suggestion.

  Our own results also indicate that $\epsilon = 0.15$ to $\epsilon = 0.5$ give reasonable results in this example. The choice $\epsilon = 0.1$ (grid spacing), however, appears to be too small as the clustering coefficient is zero in the vortex cores for this case. We will address this in paper.

- For the low-resolution case (ii), the results for $\epsilon = 0.5$ and 1.0 formerly presented in the manuscript give rise to $\rho = 0.04$ and 0.15, respectively. A maximum of $\frac{d\rho}{d\epsilon}$ is detected at about $\epsilon = 1.9$ see Figure 2, yielding $\rho = 0.45$. So, $\epsilon = 0.5$ and $\epsilon = 1.9$ appear to be reasonable choices. We have therefore substituted the right column in Figure 2 of the manuscript with the $\epsilon = 1.9$ results and added results for $\epsilon = 1.5$ ($\rho = 0.3$), as also shown in Figure 3 below. $\epsilon = 1.5$ is an interesting choice, as the average node degree $\langle d \rangle_{nn}$ appears to be "switching" here and starting to pick up regular regions instead of mixing regions as for smaller $\epsilon$.

- This study supports that the local networks measures are of course $\epsilon$-dependent, but in particular the node degree and the clustering coefficient are robust within a reasonable range of $\epsilon$-values, even for $\epsilon = 1.9$ in the low resolution case. The average node degree $\langle d \rangle_{nn}$ appears to be less robust as for increasing $\epsilon$ larger and larger parts of the network are considered for the averaging and thus the local nature of this network measure decreases. In contrast to that, the spectral results turn out to be very insensitive w.r.t the resolution of the data, the choice of $\epsilon$ and even the exact network construction. We have included a discussion on this in the revised manuscript.

3. *The methodology here proposed is really similar to the one presented by A. Hadjighasem et al. in "Spectral-clustering approach to Lagrangian vortex detection" and in some sense is an unweigthed version of their approach. The authors should explain more in detail the differences with the approach of Hadjighasem stating which are the advantages and the disadvantages of their method. An explicit numerical comparison of the two ways of building the network would significantly improve the quality of the manuscript.*

We agree that the methodology is similar. Our proposed approach is computationally cheaper, as we do not have to compute the dynamic distance between any two trajectories, but just compare the relative positions of two particles in the course of time. In particular, when two particles have come $\epsilon$-close once, we could stop considering this pair. By choosing the $\epsilon$-threshold in the proposed approach sensibly, we can ensure that the resulting network is

[Figure]

Figure 1: Network measures for high resolution initial conditions (case (i)) in the Bickley jet for $\epsilon = 0.1$ (left), $\epsilon = 0.2$ (middle) and $\epsilon = 0.5$. From top to bottom: node degrees $d$ and $\langle d \rangle_{nn}$ and clustering coefficient $C$.

[Figure]

Figure 2: Study of $\frac{d\rho}{d\epsilon}$ to determine appropriate $\epsilon$ for the Bickley jet, low resolution setting (ii).

sparse, whereas in Hadjighasem et al. post-processing steps are required to be able to compute spectral information from their resulting graph. The unweighted graph used in the present approach also allows us to use classical local network measures, whereas this would give partly blurred results when using weighted networks (see last point in response to referee # 2).

What could be done, however, is to put an $\epsilon$-threshold on the dynamic distance (instead of the minimum distance at any time step) and consider the network obtained by this construction. We will briefly address this in the conclusion.

We have chosen exactly the same parameters for the Bickley jet as in Hadjighasem et al., which allows us to directly compare the numerical results obtained by the two approaches. We therefore included a discussion in the example section 4.1.

4. *In general, the authors do not introduce correctly their work in the field of Network Theory*

[Figure]

Figure 3: Network measures for 1,000 random initial conditions (case (ii)) in the Bickley jet for $\epsilon = 0.5$ (left), $\epsilon = 1.5$ (middle), and $\epsilon = 1.9$ (right). From top to bottom: node degrees $d$ and $\langle d \rangle_{nn}$ and clustering coefficient $C$.

*application to Geophysical Systems. Climate Networks are not even cited (e.g. the seminal paper by J.F. Donges et al. "The backbone of the climate network" ) and Recurrence Networks are cited superficially without exploring the connections of the method here proposed with the Dynamical Systems Theory and the "recurrence" concept. Moreover, the authors do not confront their methodology with other network approaches previously proposed for the study of Lagrangian transport in flow systems (e.g. E. Ser-Giacomi et al. "Flow networks: A characterization of geophysical fluid transport" and M. Lindner et al. "Spatio-temporal organization of dynamics in a two- dimensional periodically driven vortex flow: a Lagrangian flow network perspective") where the local measure that they used were studied for the first time. In summary, the paper presents not only a lack of citations but also a weak framing of its scientific contents with respect to other previous works. The introduction and the Discussion and conclusions sections require also to be revised in the light of the above.*

The referee is completely right above this. We should mention that we had previously included a sentence ["We note that also the discretized transfer operator has been viewed and treated as a network, see e.g. Dellnitz and Preis (2003); Dellnitz et al. (2005); Padberg et al. (2009); Lindner and Donner (2017); Ser-Giacomi et al. (2015)."] which was mysteriously missing from the submitted version. However, we fixed this before we uploaded the online version. For the revised manuscript we have included more background in the introduction (also on Donges et al), in the final section as well as in section 3.

5. *The local measures proposed in the paper i.e. degree and clustering coefficient show an interesting detection power for mixing and no-mixing regions. However, this point is not well developed from a theoretical point of view. Which are, for instance, the quantitative relations among the degree centrality in this kind of networks and the classical Lagrangian diagnostics? Why the clustering coefficient seems to detect region of weak mixing? Maybe it could be related to some retention propriety of the flow? What is the meaning of a spectral analysis performed on a adjacency matrix that has been built thresholding Lagrangian trajectories distances at different times? Addressing these questions would certainly increase the impact of this paper, making possible connections with more theoretical approaches to Dynamical Systems.*

We have included some discussion on the relation of node degree to finite-time Lyapunov exponents, which is a widely used indicator of stretching behavior in flows. We have also addressed why the clustering coefficient detects regular, i.e. non-mixing behavior.

Our network encodes how material is transported in the flow – in space and time. Coherent regions are characterized by low fluid exchange with the surrounding flow. A spectral analysis in terms of a balanced cut problem exactly determines almost decoupled subgraphs, which correspond to dynamically "isolated" bundles of trajectories that make up coherent sets. We will provide more details in section 2 of the revised manuscript.

6. *Abstract (line 5): not clear, "Lagrangian trajectory data" is a vague definition. Please, specify that the author refer to low-resolution or incomplete data.*

   We have fixed this.

7. *Abstract (line 10): average neighbors degree and clustering coefficient are missing from the list of local network measures, please complete.*

   We have included these.

8. *Introduction (line 26): the author should compare here their approach with the work by A. Hadjighasem et al. here.*

   We have included a comparison as outlined above.

9. *Introduction (line 31-32): node degree and clustering coefficient have been already studied in the context of Lagrangian transport (E. Ser-Giacomi et al. "Flow networks: A characterization of geophysical fluid transport" and M. Lindner et al. "Spatio-temporal organization of dynamics in a two-dimensional periodically driven vortex flow: a Lagrangian flow network perspective" and V. Rodriguez-Mendez "Clustering coefficient and periodic orbits in flow networks"). The authors should discuss differences and similarities of their way to compute these metrics comparing with the ones defined previously in the literature.*

   We included more discussion on previous work in the introduction.

10. *Network of Lagrangian flow trajectories (line 9): please reformulate, the expression "network of these Lagrangian trajectories" is confusing.*

    We have rephrased this.

11. *Local network measures (line 17): "immediate importance" is not a proper definition for the degree centrality.*

    We have deleted the statement.

12. *Local network measures (line 26-27): again, the authors should refer to the papers that introduced these kind of network measures previously and not only add self-citations.*

    Done.

13. *Local clustering coefficient (line 2): what the authors mean for "certain subgraph"?*

We have rephrased this and explained that the connectivity of induced subgraphs is considered.

14. *Spectral graph partitioning (line 10-13): connected components and communities are not the same thing, please reformulate.*

    The referee is right, but we have used the two terms in two different contexts: By connected components we mean the different parts of an unconnected graph. By communities, we mean nearly decoupled parts in a connected network.

15. *Bickley jet (line 16-17): as wrote in major points: a study of robustness is needed. See: "Ambiguities in recurrence-based complex network representations of time series" by R. Donner et al.*

    We have done that (see above).

16. *Bickley jet (line 23-24): again, the strong sensitivity of the clustering coefficient to the choice of the threshold demonstrates that a deeper study is needed to address the robustness issue. Note that the clustering pattern is almost inverted just switching from $\epsilon = 0.1$ to $\epsilon = 0.2$.*

17. *Bickley jet (line 29): It is interesting that the clustering seems to be more robust in the low-resolution case, the authors could discuss this point and try to explain the reason.*

    The zero clustering coefficients in the vortex cores are an artifact of $\epsilon$ being the grid size, apart from that the clustering coefficient appears to be very robust; we have addressed this in the manuscript now.

18. *Stratospheric polar vortex (line 11): Here the authors use just one value of the threshold $\epsilon$. It is not sufficient. I think that, if they would perform a systematic robustness analysis for the idealized flow, here it would not needed to repeat it. But, at least, they should show the results for a few values of $\epsilon$, possibly selected accordingly to the indications from the robustness analysis for the Bickley jet.*

19. *Stratospheric polar vortex (line 12-14): The authors, for consistency, should show the results of the calculation of the clustering coefficient also for this more realistic case.*

    We have extended the polar vortex study to include a proper discussion on the choice of parameters as well as two different time windows. For consistency we will also show results for the clustering coefficient.

**Referee # 2**

1. *The paper entitled "Network-based study of Lagrangian transport and mixing" by Kathrin Padberg-Gehle and Chrisiane Schneide presents a method for identifying mixing regions and coherent sets in fluid flows. It constructs an undirected and unweighted network among trajectories of fluid elements: links represent at least one close encounter between the corresponding trajectories. Classical network measures and graph partitioning schemes are then applied. The method is tested numerically in 2D flows with pronounced vortical features: in a conceptual geophysical model and in atmospheric reanalysis data. The method introduced in this paper*

*is novel (in particular, it is essentially different from other network approaches that have appeared so far in the literature of fluid dynamics), and it may represent a significant technical advancement in the topic of mixing and coherence in fluid flows. The assumptions underlying the method are plausible, and the results of the numerical tests are convincing about its performance. Its main advantages are its low computational cost and its applicability for sparse or incomplete trajectory data. The paper is well-written and relatively short, and I find the very concise formulation to be an advantage. There are a few points, however, where giving further details or providing a longer, more general discussion would be useful or even necessary. In certain cases, even further numerical work may be needed. Based on the above aspects, I definitely recommend to accept the paper, with certain amendments I list below.*

We thank the referee for their careful reading of the manuscript and for their very helpful comments and suggestions. We address these below.

2. *An important aspect is the dependence of the constructed network and of the extracted features on the time interval considered. This issue should be discussed. I am thinking of some discussion similar to those in Sections I.b. and V. of Rypina and Pratt (2016). Furthermore, it might also be explained shortly for the example of Bickley jet what would happen if shorter or longer intervals of time were taken. What is more important, however, is a deeper analysis of the case of the stratospheric polar vortex from this point of view. In particular, after one of the two vortices dissolves, the particles formerly composing the dissolving vortex are dispersed within the "background sea", i.e., the larger region around the polar vortex that constitutes a different flow regime. As a result, this "background sea" is "polluted" by light yellow particles in the right plots of Figs. 7 and 8. It should be explained (and maybe numerically confirmed) that this would not happen if the network were constructed from trajectory data only from late October. I would also be happy to see the plots when the network is constructed from trajectory data from 1st to 26th September: maybe the polar vortex is delineated more clearly with this choice, with less green colour seen in the "background" regime. Not directly related to these considerations, I would be interested as well to know if the dissolution itself of one of the two vortices can explicitly be detected by some network measure (along with some appropriate choice of the time interval)?*

For the case of the Bickley jet, we prepared results on the local network measures with $\epsilon = 0.2$ and very short and very long time intervals of $[10, 14]$ and $[10, 50]$, see Figure 4. Like any other finite-time approach, the results depend on the time interval under consideration, but for the case of the Bickley jet (with the chosen parameters) the results do not change much, unless very short intervals are considered. We have added a respective statement in section 4.1, but we would rather not display any more results in the manuscript (unless the referee and editor insist on this), as in our opinion the time interval under investigation is always fixed by the user or by the available data.

For the polar vortex case, the second eigenfunction identifies the coherent set more sharply when only the time span until September 26 is considered, see Figure 5 below. We will include this analysis in the paper (including local network measures for the two time spans.).

Indeed, it would be interesting to see a precursor of the bifurcation in some network measures but this is subject to future work. We have added a little statement in section 4.2.

[Figure]

Figure 4: Node degrees $d$, $\langle d \rangle_{nn}$ and clustering coefficient $C$ (top to bottom) for time intervals [10,14] (left) and [10,50] (right).

3. *The relation of the network measures applied in this paper to the trajectory encounter number of Rypina and Pratt (2016) is described on lines 30-32 of page 2, 24-26 of page 4, and 20-21 of page 11 as "capturing very similar information", "being very related", or that "there is an obvious link". In my understanding, the trajectory encounter number of Rypina and Pratt (2016) is exactly the same as the node degree $d_i$ introduced in Section 3.2 of the present paper. The difference is that the definition here is expressed in a more formal way, and, more importantly, this quantifier is placed here in a much wider context (that of the trajectory-based flow network). I think that this has to be formulated clearly at each occurrence (enumerated above), and references to other network measures (average degree of neighbouring nodes, local clustering coefficient) have to be completely avoided in this relation.*

   We have rephrased these statements.

4. *Page 4, line 4: "By construction": It would be better to write: "By our choice". In the same line: "so there are": It should be "since there are" or "i.e., there are".*

   Changed as suggested.

5. *Page 5, line 13: A period is missing after the parentheses.*

   Fixed.

6. *Page 5, lines 15-16: A reference should be included for the statement.*

   Done.

[Figure]

Figure 5: Second eigenfunction of the polar vortex network on September 1, 2002 (left) and September 26, 2002 (right). Top row: trajectories from September 1 - October 31, 2002 have been considered for setting up the network. Bottom row: only trajectories from September 1 - 26, 2002 have been considered for setting up the network.

[Figure]

Figure 6: Node degrees $d$ (left), $\langle d \rangle_{nn}$ (middle) and clustering coefficient $C$ (right) for temporal resolution of the trajectory data of 0.1 (instead of 1.0 in the manuscript).

7. *Page 5, line 25: Maybe a reference can be included for the k-means as well.*

   We included a classical reference for the k-means algorithm.

8. *At the beginning of Section 4.1: I understand that the Bickley jet is just an arbitrary example, and the main motivation for choosing it was, I guess, the presence of results in the literature that are comparable to those in the current paper. Still, it would be useful to mention the geophysical relevance of the Bickley jet (especially its relation to the other example: the stratospheric polar vortex), and the main properties of the flow with the current parameter setting (like the presence of a meandering central jet and of three regular vortices on each side).*

   We added a short remark addressing these points at the beginning of section 4.1.

9. *Page 6, line 10, and also page 10, line 10: "we output the particle positions ...": Are only these positions considered for the network construction? If so, it should be explicitly stated (and this raises further, maybe crucial questions about the dependence on the choice of the output). If not (as I suppose), where is advantage made of this special filtering of the trajectories in the paper? (For e.g. the interpretation of the figures, informing the reader about this choice of outputting is not needed.)*

   We illustrate that the choice of temporal resolution for the output of particle positions does not determine the outcome of the analysis by choosing time steps of 0.1 (instead of 1.0) for the high-resolution case of the Bickley jet with $\epsilon = 0.2$ and the time interval $[10, 30]$. Results are given in Figure 6. We have added a small comment in section 4.1 that the results are insensitive to the temporal resolution in this example. Of course, a coarse temporal resolution sparsifies the trajectory data and is therefore always computationally advantageous.

10. *Page 6, lines 21 and 28: "Spurious" is not an appropriate word here (it approximately means "false"). "Fuzzy" may be a good substitution.*

    We have substituted "spurious" by "fuzzy".

11. *Page 6, line 24: What does "singularity" mean in the present context? Some more explanation should be given.*

    We have changed the wording and added an explanation that due to the small $\epsilon$ (= grid size) no triangles are found in the regular region for $\epsilon = 0.1$.

12. *Page 8, line 5: "four eigenvalues" should be "three eigenvalues".*

We have changed this.

13. *Figure 6: It might be mentioned very shortly that the clustering finds a few false yellow points for the sparse case.*

    We have added a respective statement in the text.

14. *Page 11, line 14: After the expression "sparse data", it may be emphasized that this is illustrated by case (ii) in the Bickley jet investigation.*

    We have added a respective statement.

15. *I believe that the applicability of the proposed method is not restricted to volume-preserving flows. For example, this method might be especially useful in open contracting dynamics. It might be beneficial to mention this in the paper, and to indicate that the numerical examples discussed in this paper originate from volume-preserving flows.*

    Indeed, the method is not restricted to volume-preserving flows and we have added a respective statement in the discussion/conclusion section.

16. *I think that choosing epsilon such that the graph is not connected may also give insightful information in certain cases, especially in non-volume-preserving dynamics, which could be mentioned. Anyway, it should be more clearly stated in line 16 of page 3 that the present paper considers choices of epsilon only that give a connected graph.*

    We have included a remark at the beginning of section 2 that we only consider connected networks. In the discussion/conclusion section we mention that unconnected networks might also be insightful to consider.

17. *Would the authors consider to be a useful possibility to count the total number of encounters between trajectories occurring in the course of time, and to construct a weighted network based on these numbers? If so, it might be mentioned in Section 5, as a natural extension to the presently proposed method.*

    This is a very good point and we set up such as "weighted" network and analyzed it. For the Bickley jet example this construction leads to blurred results for the node degrees due to the accumulation of encounters also in low-mixing regions. In the clustering coefficient, high- and low-mixing regions are clearly distinguishable. The extraction of coherent sets from the eigenvectors of the generalized graph Laplacian eigenvalue problem appears undisturbed. See Figure 7 below.

    In the manuscript, we have added a short discussion describing these results in the final section (Discussion and conclusion).

[Figure]

Figure 7: Node degrees $d$, $\langle d \rangle_{nn}$ (top) and clustering coefficient $C$ as well as coherent sets for the weighted network in case of the Bickley jet (bottom).

---

## Author Response (AR1)

**Authors' response**

We thank the referees for their careful reading of the manuscript and their very insightful and constructive comments and suggestions. We have outlined all major changes in the detailed response to the referee reports at

http://www.nonlin-processes-geophys-discuss.net/npg-2017-4/npg-2017-4-AC1-supplement.pdf

Here we include a marked-up version of the manuscript highlighting all changes made.

[revised manuscript text omitted]

---

## Author Response (AR2)

**Authors' response**

We thank the Gábor Drótos for his careful reading of the revised manuscript and the constructive comments and suggestions, which we have fully considered in our revision. We reply to all points below and also include a marked-up version of the manuscript highlighting all changes made.

1. *The authors state in their response to Referee #1, in point 5, that they will provide more details about the spectral analysis in section 2. However, I did not find this part in the paper, neither in section 2, nor in section 3.3 (where it would fit better). (This discussion is, however, already part of the response, so it just needs to be adapted to the text of the paper, and included at an appropriate position.)*

   We have included the following paragraph at the beginning of sect 3.3, clarifying this point:

   "Spectral graph partitioning aims at decomposing a network into components with specific properties. In our setting, the network encodes how material is transported by the flow, both in space and time. We are interested in identifying coherent structures in the flow, which are known to be organizers of fluid transport. From a spatio-temporal point of view, coherent sets are formed by trajectories that stay close to each other (Froyland and Padberg-Gehle (2015)) and thus are more tightly connected than others. Such information can be obtained by solving a balanced cut problem of the network (Hadjighasem et al. (2016)). "

2. *Page 7, lines 25-26: From this discussion, a conclusion should be explicitly drawn about the reasonable choice for the investigated range of epsilon.*

   We have adapted the description as follows:

   "A maximum of $\frac{d\rho}{d\epsilon}$ is detected at about $\epsilon = 1.9$, yielding $\rho = 0.45$, which already corresponds to a dense network. So a reasonable range appears to be $\epsilon \in [0.5, 1.9]$. "

3. *Page 8, lines 9-16: It should be mentioned that the jet core is not resolved by any of the local network measures in the considered (sparse) setting.*

   Done.

4. *In the same paragraph: It should be explained why "blurring the local information" (with increasing epsilon) results in increased average node degree $\langle d \rangle_n n$ in the regular regions compared to the mixing regions.*

   We have adapted the explanation:

   "This is again due to the enlarged neighborhood, where averages are now crucially influenced by flow regimes outside the local neighborhood of the trajectory under consideration. For instance, for a node with a small node degree, its neighborhood extends far into the mixing regions characterized by large node degrees, resulting in a large average degree for this node (and vice versa for nodes with large node degree)."

5. *Page 12, lines 10-11: It would be beneficial to extend the sentence "The local clustering coefficient is large in particular in the core of the vortex" by adding: ", where the node degree and the average node degree take on smaller values", or something similar.*

Changed as proposed by the referee.

6. *Page 12, lines 15-16: The phrasing "with isolated yellow regions in the background flow" might need to be clarified, e.g. like "with small, isolated yellow regions dispersed in the background flow".*

Changed as proposed by the referee.

7. *Finally, responding to point 2 of the authors' response to my earlier comment, I agree that more numerical results should not be displayed for the Bickley jet. At the same time, I emphasize again that the dependence on the time interval carries useful information about the inherent properties of the system's dynamics (as illustrated by the case of the stratospheric polar vortex), and should not be "fixed by the user or by the available data". (If only restricted data is available, subintervals can be considered.)*

The referee is right and we have included a statement at the end of section 2:

[revised manuscript text omitted]